**Improved Snow Property Retrievals by Solving for Topography in the Inversion of At-sensor Radiance Measurements**

Brenton A. Wilder[1], Joachim Meyer[1], Josh Enterkine[1], Nancy F. Glenn[1*]

[1]Department of Geosciences, Boise State University, Boise, ID, USA

Correspondence to: Nancy F. Glenn (nancyglenn@boisestate.edu)

**Abstract**

Accurately modelling optical snow properties like snow albedo and specific surface area (SSA) are essential for monitoring the cryosphere in a changing climate and are parameters that inform hydrologic and climate models. These snow surface properties can be modelled from spaceborne imaging spectroscopy measurements but rely on Digital Elevation Models (DEMs) of relatively coarse spatial scales (e.g. Copernicus at 30 m), which degrade accuracy due to errors in derived products – such as slope and aspect. In addition, snow deposition and redistribution can change the apparent topography and thereby static DEMs may not be considered coincident with the imaging spectroscopy dataset. Testing in three different snow climates (tundra, maritime, alpine), we established a new method that simultaneously solves snow, atmospheric, and terrain parameters, enabling a solution that is

more unified across sensors and introduces fewer sources of uncertainty. We leveraged imaging spectroscopy data from AVIRIS-NG and PRISMA (collected within 1 hour) to validate this method and showed a 25% increase in performance for the radiance-based method over the static method when estimating SSA. This concept can be implemented in missions such as Surface Biology and Geology (SBG), Environmental Mapping and Analysis Program (EnMap), and Copernicus Hyperspectral Imaging Mission for the Environment (CHIME).

**Key Words:** Imaging Spectroscopy, Snow Properties, Topography, Snow Albedo

**1 Introduction**

Accurately mapping snow surface properties is essential for seasonal snow zones in a changing climate especially in regions where seasonal snowpack is expected to change dramatically in the coming decades (Siirila-Woodburn et al., 2021). For example, snow albedo plays a crucial role in melting of the snowpack during the ablation season (Wang et al., 2020) with changes in snow albedo directly affecting the amount of absorbed solar radiation, and therefore the amount of snow that is melted off. Throughout the winter season, snow albedo fluctuates due in part to grain size (Seidel et al., 2016) and light absorbing particles (Kaspari et al., 2015; McKenzie, 2020; Schmale et al., 2017; Skiles & Painter,

2017). With a limited number of *in situ* snow stations around the globe, and the snow surface
constantly undergoing metamorphism across space and time, satellite imagery represents the
best potential for spatially and temporally complete mapping of snow properties. Accurately
retrieving snow albedo and other snow surface properties from satellite imagery is
paramount, especially in a rapidly changing climate (Malmros et al., 2018).

Retrieval of snow properties from satellite remote sensing relies on Digital Elevation

Models (DEMs) to correct for local terrain effects (Bair et al., 2021; Bair et al., 2022; Dozier
et al., 2022). In a previous study, researchers found global DEM products to have "blunders
and errors" when compared to airborne lidar, particularly in derived slope and aspect which
cause severe errors in calculated cosine of local solar illumination angles ($\mu_s$) (Dozier et al.,
2022). They found errors in $\mu_s$ ranging from 0.048 to 0.117 (dimensionless) across several
sites for Copernicus global DEMs caused by errors in slope and aspect. The $\mu_s$ term is a
function (Eq. 1) of slope angle (S), slope azimuth angle or aspect (A), solar zenith angle ($\theta_0$),
and solar azimuth angle ($\phi_0$) – with the last two being well constrained:

$$\mu_s = \max[0, \ \cos(\theta_0)\cos(S) + \sin(\theta_0)\sin(S)\cos(\phi_0 - A)] \quad (1)$$

Because $\theta_0$ and $\phi_0$ are calculable with low errors, the biggest contribution to errors in $\mu_s$
stem from slope and aspect. Errors in $\mu_s$ increase monotonically with increasing $\theta_0$ (e.g., sun
setting has high $\theta_0$, as does solar noon in high latitude winters). This phenomenon can be
explained by plotting Eq. 1 for various $\theta_0$ (Figure 1). Put simply, at higher $\theta_0$ there is a
higher standard deviation in $\mu_s$ surrounding a known slope and aspect (with some temporally
consistent uncertainty), increasing the probability and magnitude of such an error. If one were
to compute standard deviations of $\mu_s$ across varying $\theta_0$, one would arrive at similar errors of
$\mu_s$ presented in Dozier et al. (2022). For clarity, in Figure 1 we have highlighted an example
case with slope=25° +/- 4.73 and aspect=280° +/- 36.3. Example uncertainties for this
exercise can be found in Table 2 of Dozier et al. (2022).

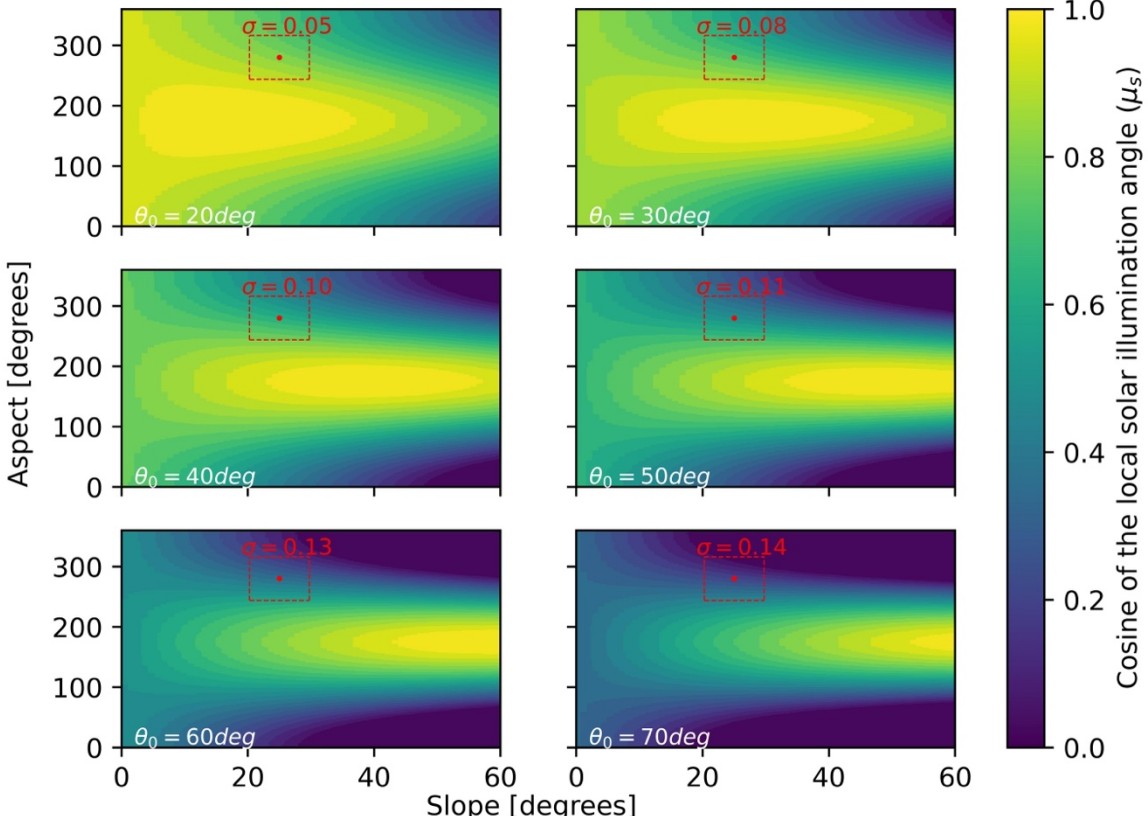

**Figure 1.** Cosine of local illumination angles ($\mu_s$) as a function of slope (x-axis) and aspect (y-axis) incremented by 1°, illustrating the problem at higher latitude, and/or winter acquisitions, where standard deviation ($\sigma$) of $\mu_s$ increases monotonically with solar zenith angles ($\theta_0$). Aspect is shown here measured clockwise from north (with north containing a discontinuity at 360 degrees). For this illustration $\phi_0$ is fixed at a value of 175°. The red dots represent the example point at slope=25° +/- 4.73 and aspect=280° +/- 36.3 and are bordered by their uncertainty and the resulting $\sigma$.


Recent work has shown $\mu_s$ can be modelled using an optimal estimation framework
given the Top of Atmosphere (TOA) radiance observed from imaging spectroscopy (Carmon
et al., 2023). The authors solve for surface, atmospheric, and topographic state variables
simultaneously in their model. This works physically because the partition of direct to diffuse
light introduces a shape and magnitude effect on the TOA radiance spectra. However,
retrieving snow optical properties is sensitive to directional reflectance which is significantly
influenced by the viewing geometry and surface roughness (Bair et al., 2022), leading to
possible shortcomings in this method specifically for snow covered pixels. To address this
and expand upon this framework, we present a new method to account for terrain in snow
covered areas. Our method was tested on pixels with greater than 75% snow cover in three
different snow climates (tundra, maritime, and alpine) with spaceborne imaging spectroscopy
with the aim to reduce error in derived snow properties by optimally solving for topography.
The spaceborne results are validated against high confidence airborne spectrometer data. This
work directly contributes to snow property retrievals in steep terrain and/or at times of high
solar zenith angles for satellite imaging spectroscopy missions such as Surface Biology and
Geology (SBG) (Cawse-Nicholson et al., 2021), Copernicus Hyperspectral Imaging Mission
for the Environment (CHIME) (Celesti et al., 2022), and EnMap (Guanter et al., 2015).

## 2 Methods

## 2.1 Study area

For our study, we used *PRecursore IperSpettrale della Missione Applicativa* (PRISMA) imagery over three sites capturing different snow climates and solar zenith angles: San Juan Mountains (Colorado, USA, 29 April 2021, $\theta_0$=27°), Mount Shasta (California, USA, 28 February 2021, $\theta_0$=52°), and the Toolik area (Alaska, USA, 21 March 2021, $\theta_0$=68°) (Figure 2). The San Juan Mountains location is considered a high alpine site located in interior continental USA with an elevation range of 2208-4129 m. The Mount Shasta site is a maritime snow climate along the western coast of USA with an elevation range of 750-4232 m. The Toolik site (elevation range = 504-1748 m) is a high-latitude tundra site, being mostly flat but with steep sections along the Brooks Range (along the southern part of the image). PRISMA, launched by the Italian Space Agency (ASI) and beginning operation on March 22, 2019, is a spaceborne imaging spectroscopy mission collecting radiance at 30 m spatial resolution across 239 bands spanning 400-2500 nm at a spectral resolution better than 12 nm across the visible-near and shortwave infrared (Cogliati et al., 2021).

To validate our method, we used four existing Airborne Visible Infrared Imaging Spectrometer-Next Generation (AVIRIS-NG) flightlines over the San Juan Mountains from 29 April 2021 (flying 1 hour after PRISMA acquisition). AVIRIS-NG collects radiance measurements at variable spatial resolution (depending on the flight altitude) across 425

bands spanning 380-2510 nm in 5nm intervals (Green et al., 2023). For this flight, data were
collected at 4 m spatial resolution. We downloaded AVIRIS-NG apparent reflectance from
National Snow and Ice Data Center (NSIDC) and observation geometry data from NASA
Search Earth Data (Skiles & Vuyovich, 2023).

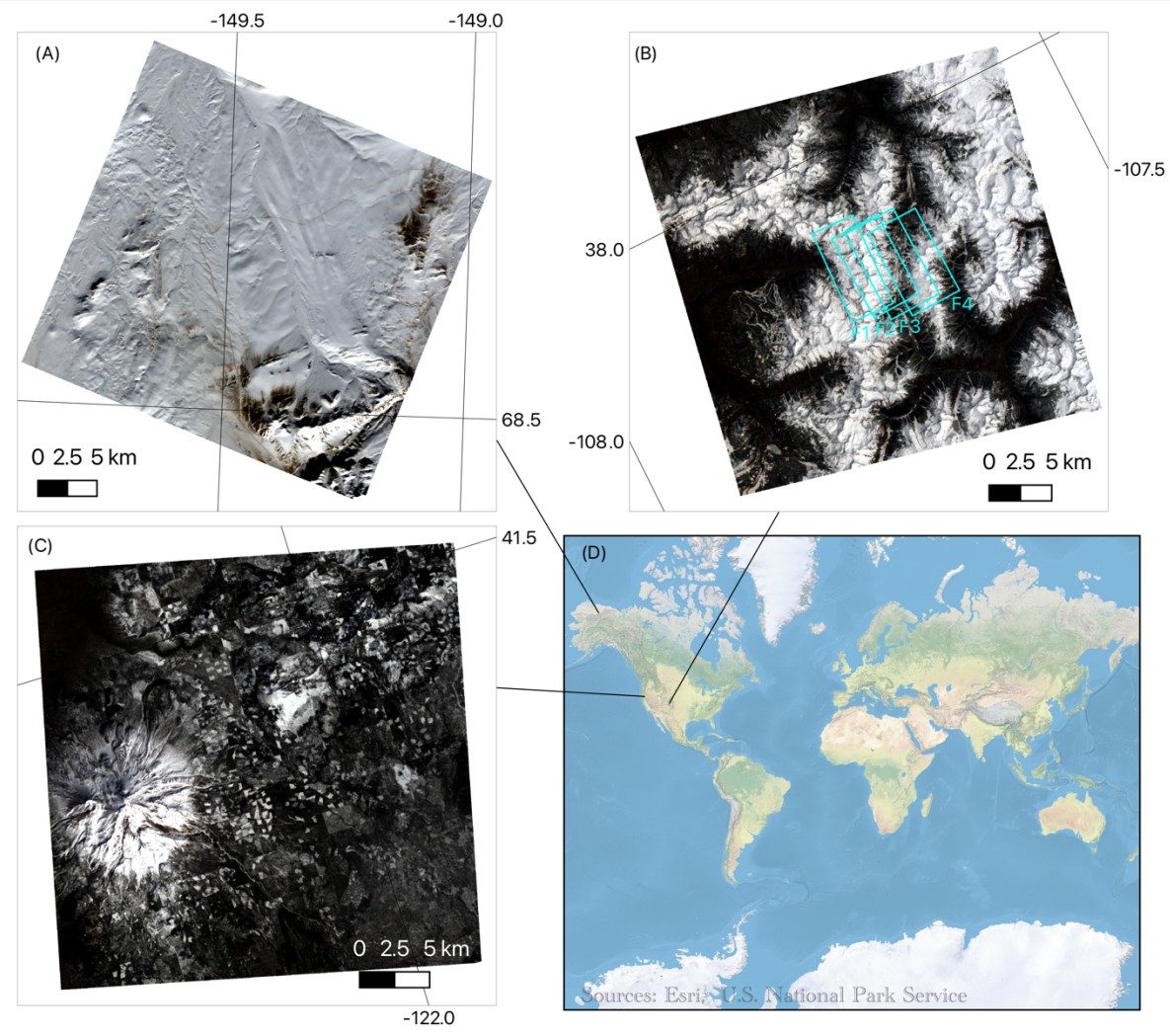


**Figure 2.** PRISMA true colour images for Toolik on 21 March 2021 (A), San Juan Mountains on 29 April 2021 (B), and Mount Shasta on 28 February 2021 (C). Four coincident AVIRIS-NG flightlines (F1-F4) are shown in cyan over the San Juan Mountains.

## 2.2 Modelling surface, atmosphere, and topography from PRISMA

The algorithmic improvements build off a workflow that estimates snow properties given PRISMA TOA radiance, titled Global Optical Snow properties via High-speed Algorithm using K-means (GOSHAWK) (Wilder et al., 2024). In short, our method uses the analytic asymptotic radiative transfer model (AART) (Kokhanovsky & Zege, 2004) coupled with libRadtran (Mayer & Kylling, 2005) to invert snow surface and atmospheric properties (Bohn et al., 2021; Dalcin & Fang, 2021), and fractional covers of mixed pixels under varied lighting conditions using non-linear numerical optimization (Bair et al., 2021). The parameters solved for in the optimization routine include fractional covers, specific surface area (SSA), light absorbing particle concentration (modelled as dust), liquid water percentage, dimensionless aerosol optical depth at 550nm, and columnar water vapor in the atmosphere. Here, we expand upon the algorithm considering recent work showing the capacity to estimate $\mu_s$ from TOA radiance (Carmon et al., 2023; Bohn et al. 2024). This idea is demonstrated in Figure 3 using fixed snow properties via AART and fixed atmosphere properties via libRadtran across the range of plausible $\mu_s$ (i.e. 0 to 1). Like the findings in

Carmon et al. (2023), Figure 3 shows that $\mu_s$ controls both the spectral shape and magnitude
of observed TOA radiance with the effect varying across wavelengths. The greatest shape
effect can be seen in the visible spectrum (roughly 400-700 nm) due to the magnitude of the
diffuse irradiance. In combination with the magnitude and shape shift, this parameter
becomes solvable during optimization due to its strong separability – especially when
considering the entire spectrum data from a hyperspectral remote sensing source such as
PRISMA. It is important to note that $\mu_s$ impacts both the AART estimation of snow
reflectance and libRadtran estimation of incoming solar irradiance.


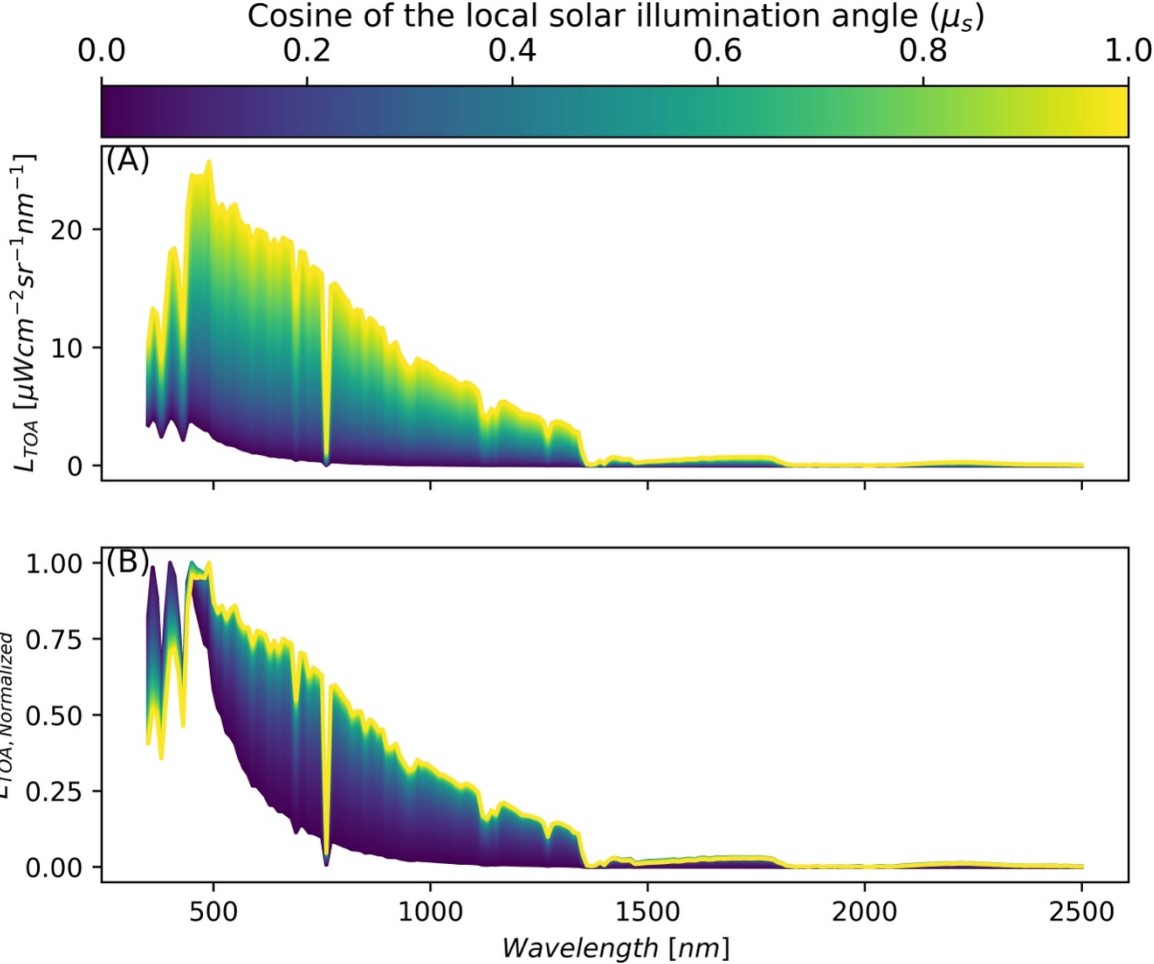


**Figure 3.** Synthetic data showing change in magnitude (A) and shape (B) of top of

atmosphere radiance ($L_{TOA}$) with respect to changing local solar illumination angle ($\mu_s$) for

fixed snow surface state variables modelled with AART, and fixed atmospheric state

variables modelled with libRadtran (viewing geometry was fixed as well). State variables and

solar/view geometry were based on a PRISMA acquisition over southern Idaho on 8

December 2022. Figure (B) shows normalized radiance with respect to peak TOA radiance
across wavelengths to highlight the change in shape.

However, if we were only to optimize $\mu_s$, the other key terms, local viewer zenith

angle ($\mu_v$) and local phase angle ($\xi$) in the AART formulation for bidirectional reflectance of
snow (Eq. 2) (Kokhanovsky & Zege, 2004; Kokhanovsky et al., 2021a) would remain
constant from the available DEM (i.e., $\mu_s, \mu_v, \xi$ are all derived from DEM),

$$r_{snow}(\mu_s, \mu_v, \xi, \lambda) = r0(\mu_s, \mu_v, \xi) \ a_{snow}(\lambda)^f \qquad (2)$$

where $r0$ is the reflection function of a semi-infinite non-absorbing snow layer (Tedesco &
Kokhanovsky, 2007), $\alpha_{snow}$ is the spherical albedo [plane albedo can be computed using (26)
in Kokhanovsky et al. (2021a)], f is the escape function (Kokhanovsky et al., 2021a), and
$r_{snow}$ is the bidirectional reflectance of snow. Keeping other terms $\mu_v \ and \ \xi$ the same are
problematic because snow reflectance is poorly approximated as a non-Lambertian surface
(Leroux & Fily, 1988), and the outcome will be greatly influenced by $\mu_v \ and \ \xi$. Therefore, to
incorporate solving for $\mu_s, \mu_v, \ and \ \xi$ from TOA radiance into the algorithm, we instead elect
to optimally solve for cos(aspect) (i.e., "northness") and sin(aspect) (i.e., "eastness") (Table

1).


**Table 1.** Parameter space and initial vectors used in numerical optimization for PRISMA

data.

| Parameter [unit] | Definition | Feasible Range | Initial State | Type |
|---|---|---|---|---|
| $f_{snow}$ [%] | Fractional snow in the mixed pixel | [0, 100] | 10 | Surface |
| $f_{shade}$ [%] | Fractional shade in the mixed pixel | [0, 100] | 20 | Surface |
| $f_{LC1}$ [%] | Fractional cover of endmember 1 (based on land cover value at pixel) | [0, 100] | 50 | Surface |
| $f_{LC2}$ [%] | Fractional cover of endmember 2 (based on land cover value at pixel) | [0, 100] | 20 | Surface |
| SSA [$m^2$ $kg^{-1}$] | Specific surface area (SSA) | [2, 156] | 40 | Surface |
| LAP [$\mu g$ $g^{-1}$] | Concentration of light absorbing particles, LAP, modelled as dust (PM-2.5). | [0, 145] | 0 | Surface |
| Liquid water [%] | Percentage of liquid water on the snow surface | [0, 50] | 2 | Surface |
| AOD 550 [%] | Dimensionless Aerosol Optical Depth (AOD) at 550 nm | [1,100] | 10 | Atmospheric |
| $H_2O$ [mm] | Columnar water vapor in the atmosphere | [1,50] | 1 | Atmospheric |
| Eastness | sin(aspect) | [-1,1] | Variable | Topographic |
| Northness | cos(aspect) | [-1,1] | Variable | Topographic |


Aspect can be solved during optimization by using the atan2 function. We chose to use this

method because eastness and northness are continuously differentiable, and therefore, are

suited for numerical optimization methods, whereas aspect is discontinuous at north (using

the convention of 0 and 360 degrees as north). We then can use this optimal aspect to
estimate $\mu_s$ (Eq. 1), $\mu_v$, and $\xi$. This directly impacts Eq.2 and Eq. 5 (formulation of incoming
solar energy in the model) (Picard et al., 2020),

$$E(\lambda) = \psi\mu_s E(\lambda)_{dir} + V_\Omega E(\lambda)_{diff} + \left[\left(1 + \frac{\cos(S)}{2} - V_\Omega\right)r(\lambda)_{surf}E(\lambda)_{diff}\right] (5)$$

where E is total incoming irradiance, $\psi$ is binary shade or no shade, $E_{dir}$ and $E_{diff}$ are the
direct and diffuse irradiance, respectively, $V_\Omega$ is the sky view factor (Dozier, 2022), and $r_{surf}$
is the reflectance of nearby terrain (which is assumed to be equal to the pixel itself). The term
E is solved within our non-linear numerical optimization method as described in Wilder et al.
(2024). This was modelled incorrectly in Wilder et al. (2024); however, this was corrected in
this paper where only diffuse irradiance is used in the 3rd term in Eq. 5. Also, adding in the
two extra parameters (eastness and northness) in our updated optimization scheme did not
change our run time significantly. Caution is advised against solving for slope *and* aspect in
the inversion due to the non-unique solution space (Figure 1); however, only considering
aspect ensures unique solutions of aspect, $\mu_s$, $\mu_v$, and $\xi$. We chose aspect because of its
greater impact on determining partition of direct and diffuse illumination and has been found
to be more impactful to errors associated with snow property retrieval (Donahue et al., 2023).
In this study we used estimate of total ozone column as input into creating the libRadtran
look up table specific for each image. We used the average weekly ozone over the bounds of
the image from Sentinel-5P NRTI O3: Near Real-Time Ozone dataset. This approach serves
an improvement over Wilder et al. (2024), where ozone was fixed at 300 Dobson Units.

**2.3 Estimating snow properties from AVIRIS-NG for validation**
Due to the fine signal to noise ratio and spatial resolution of AVIRIS-NG, we treated
the dataset as the ground reference. It also captured a similar spectral range to PRISMA
which made it a suitable comparison dataset.  The main assumption here is that AVIRIS-NG
pixels at 4 m are relatively homogenous and are either snow or no-snow – which may not
always be the case. This could be a potential source of uncertainty in our analysis. To select
snow-covered pixels, we solved for NDSI (Normalized Difference Snow Index) using bands
at 600 nm and 1500 nm. We limited our retrieval of snow properties for NDSI greater than or
equal to 0.90 (Painter et al., 2013). A common approach to retrieve snow grain size from pure
snow pixels is to apply the scaled band area algorithm (Nolin & Dozier, 2000); however, it is
recognized that the large presence of liquid water is a limitation. The maximum air
temperature of 10.8° C on the day of the image at the San Juan Mountains site indicated that
elevated liquid water at the surface was probable (Center for Snow and Avalanche Studies,
2023). Additionally, reflectance spectra appeared to be shifted along the x-axis (wavelength)
due to the presence of liquid water. Therefore, we used constrained non-linear numerical
optimization to model apparent snow reflectance with AART by allowing fractional snow,
fractional shade, liquid water, and SSA to vary. We did not include rock or forest
endmembers in this formulation, assuming the 4 m pixels are relatively homogenous as
previously stated. Topographic incident angles were held constant based on the 4 m
resolution DEM provided by AVIRIS-NG. We minimized Root Mean Square Error (RMSE)
between observed-apparent and modelled-apparent snow reflectance from AART
wavelengths in the range, 1000-1250 nm. This range has high ice absorption and limited
impacts from atmospheric interference and LAP (Miller et al., 2016). The presence of liquid
water was included in our analysis by means of the composite refractive index of water and
ice (Donahue et al., 2022; Hale & Querry,1973; Warren & Brandt, 2008). We assumed
similar grain shape assumptions for both PRISMA and AVIRIS-NG, and that if there is a bias
due to this it should be consistent between the two datasets in our analysis.

**2.4 Comparing modelled snow properties**

The algorithm was used in two different modes: 1) static topography based on the

Copernicus DEM (hereon called "*static*"); and 2) solved topography based on the algorithm
updates (hereon called "*radiance*"). To compare the accuracy of PRISMA derived SSA and
liquid water, we resampled the AVIRIS-NG optical property results (SSA and LWC) to
match the PRISMA resolution (30 m) and extents by using bilinear interpolation. Then, we
sampled all valid pixels where PRISMA and AVIRIS-NG had snow. We then computed r-
pearson correlation coefficient, Mean Bias, and RMSE for the radiance and static methods
(with respect to AVIRIS-NG). Finally, we used Copernicus derived slope and aspect maps to
determine where the largest errors were occurring on the landscape to compare with the
theoretical basis presented in Figure 1. We do this by using the mean absolute difference with
respect to $\mu_s$ . We expected to see higher differences in north facing aspects (i.e., $\mu_s$
approaches 0), and where $\theta_0$ was higher. To test the interaction with $\theta_0$ more fully, we
extended the analysis to Mount Shasta, CA, and Toolik, Alaska, where no in *situ data*
existed. We compared the modelled properties between the radiance and static methods to
assess how these assumptions impacted results for these types of data at 30 m scale.

**2.5 Comparing DEM and radiance derived $\mu_s$**
To ensure the resulting radiance derived $\mu_s$ were valid we downloaded the best
available validation data sources for comparison. For the San Juan and Shasta sites, we
collected DEM products at 1 m spatial resolution and collected 5 m spatial resolution DEM
for the Toolik site (U.S. Geological Survey, 2019; U.S. Geological Survey, 2022). Then, we
computed slope, aspect, solar zenith angle, and solar azimuth angle for all pixels to compute
$\mu_s$ at the native resolution (Eq. 1). Then, we used bilinear interpolation to resample the 1 m
and 5 m products to 30 m to exactly match the extents and resolution of our PRISMA images.
We would like to acknowledge that while these are the best freely available datasets for our
images, they still do not capture the true snow-on topography, and instead are a
representation of the "snow-free" surface. We compared matching pixels to determine
RMSE, $R^2$, and Mean Bias. Pixels that were marked as shadow from ray tracing were
excluded from this comparison.

**3 Results**
**3.1 Validation using AVIRIS-NG data over the San Juan Mountains**
Over the area of the flightlines, AVIRIS-NG estimated mean SSA = 18.0 +/- 8.3 $m^2$ $kg^{-1}$
, PRISMA radiance method estimated mean SSA = 19.6 +/- 5.8 $m^2$ $kg^{-1}$, and PRISMA static
method estimated mean SSA = 22.0 +/- 12.1 $m^2$ $kg^{-1}$. When comparing the SSA performance
over each pixel to the AVIRIS-NG flightlines (Figure 4) we found the PRISMA radiance
method (r=0.43; RMSD=8.0 $m^2$ $kg^{-1}$; bias=+1.7 $m^2$ $kg^{-1}$; n=36,412) performed slightly better
than the static method (r=0.23; RMSD=13.6 $m^2$ $kg^{-1}$; bias=+4.0 $m^2$ $kg^{-1}$; n=36,412) for SSA.

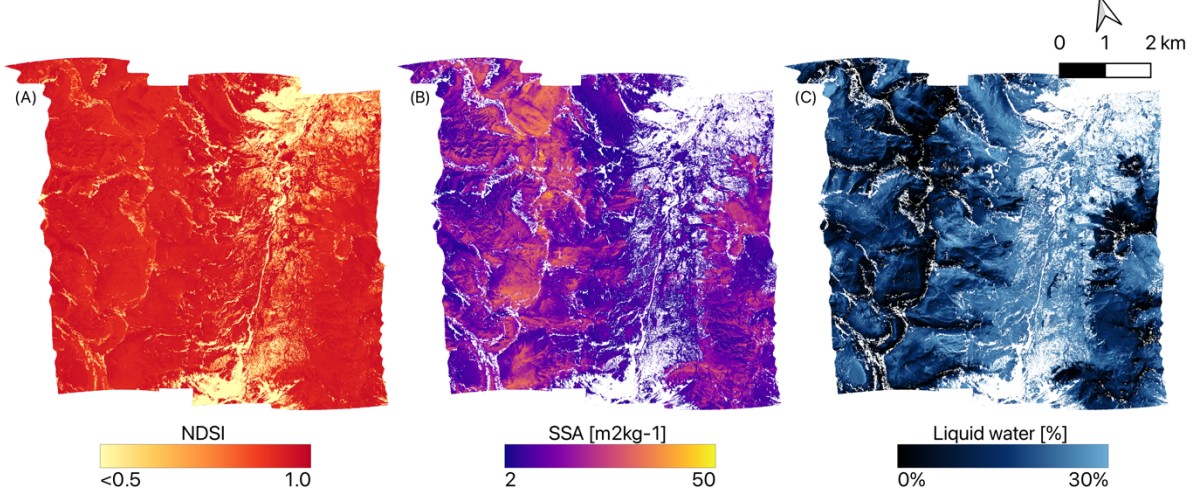

NDSI
<0.5      1.0

SSA [m2kg-1]
2      50

Liquid water [%]
0%      30%

**Figure 4.** Snow properties computed from AVIRIS-NG (4 m spatial resolution) on 29 April 2021 including NDSI (A), SSA (B), and LWC (C) for the San Juan Mountains site.

There was not a significant improvement in liquid water estimation between radiance (r=0.67; RMSD=10%; bias=-8%; n=36,412) and static (r=0.67; RMSD=10%; bias=-9%; n=36,412). Furthermore, it appeared that there was a consistent liquid water bias of 8 to 9%, hinting that more melt had occurred during the AVIRIS-NG flights. As previously noted, the temperatures were well above freezing during the overpass of AVIRIS-NG and occurred roughly 1 hour later in the day compared to the PRISMA acquisition. This most likely explains the higher liquid water and lower SSA observed by AVIRIS-NG. We further tested this by masking out areas where AVIRIS-NG liquid water content was greater than 0.1%, to establish areas where low amounts of melt occurred between the two acquisitions. We found that performance of

PRISMA static (RMSD=14.2 $m^2$ $kg^{-1}$; rRMSD=49%; n=181) and radiance (RMSD=6.9 $m^2$ $kg^{-1}$; rRMSD=23%; n=181) methods were more accurate for these areas. The radiance method
performed slightly better, suggesting a modest 25% improvement in accuracy for SSA over the
static method when considering pixels that were less impacted by melt.
Additionally, comparing all pixels we found improvement from radiance occurred
mostly on steep, north facing aspects (e.g., when $\mu_s$ approached 0). We found the absolute
residual increased as $\mu_s$ approached zero for the static method (r = -0.47; p<0.01), while this
relationship was diminished nearly by a factor of 5 for the radiance method (r = -0.10; p<0.01)
(Figure 5.A). These errors were caused by incorrect terrain information in the inversion, where
inversion error increased proportionately in the static method (Figure 5.B).

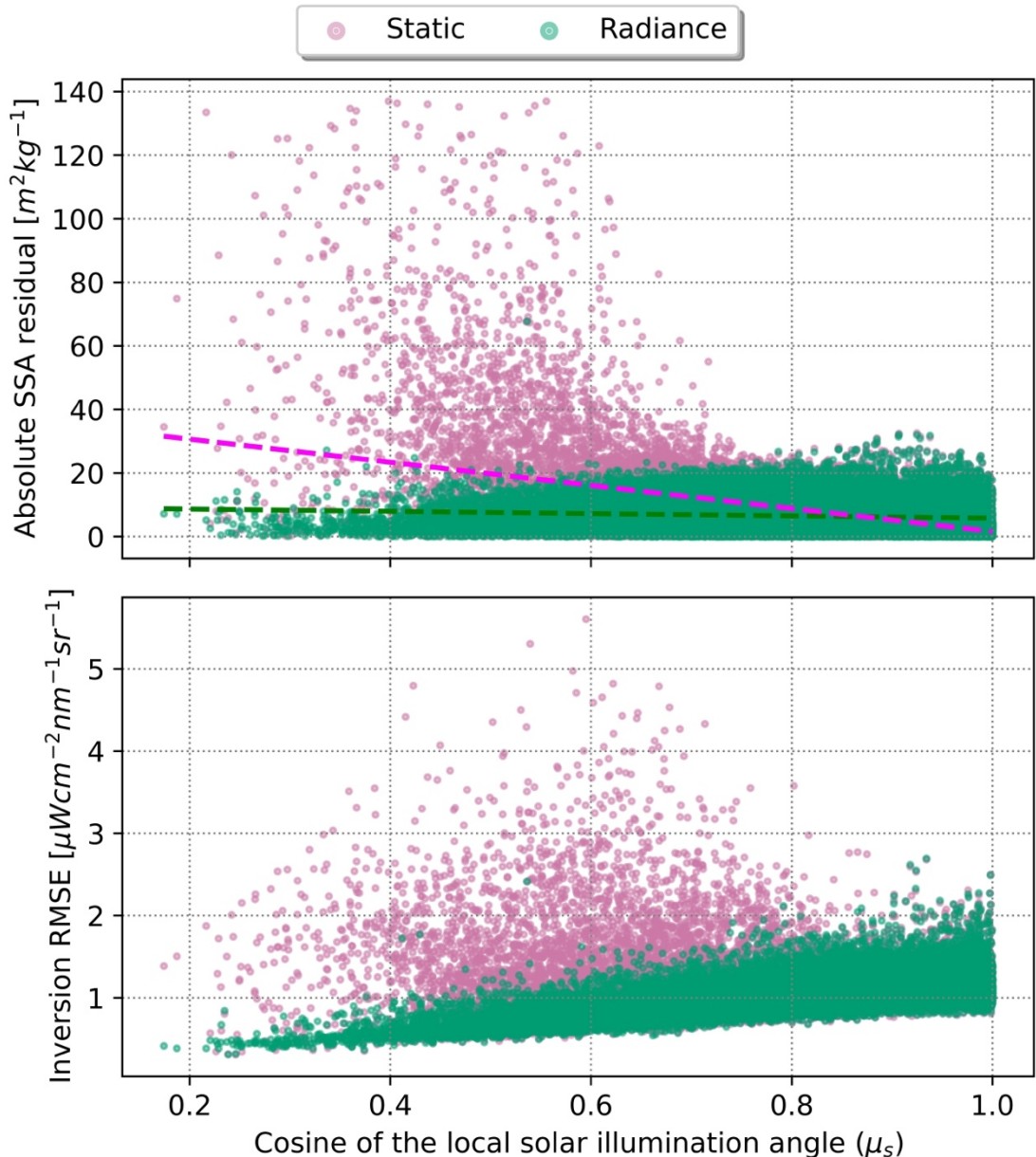


**Figure 5.** Absolute difference in modelled SSA when compared to AVIRIS-NG for radiance

method (green) and static method (pink) respect to $\mu_s$ (A) and resulting RMSE from the

inversion from PRISMA with respect to $\mu_s$ (B). Error in the static method increases
significantly when $\mu_s$ approached zero (r = -0.47; p<0.01); however, the difference was less
noticeable in the radiance method (r = -0.10; p<0.01).

**3.2 Comparing radiance and static methods between sites**
On average across each of the images, radiance and static methods provided similar
retrieved parameters within less than one standard deviation (Table 2). In general, this means
there is not a significant difference at the 30 m scale for computing parameters such as SSA
and broadband albedo (BA) when considering the entire image. Interestingly when terrain is
fixed, the static model compensated for incorrect illumination by increasing the aerosol optical
depth (thereby reducing the amount of direct solar radiation). Investigating the errors more
closely, we found much larger differences in retrieved properties where $\mu_s$ approached 0
(Figure 6). The difference in distributions matched closely to the theoretical demonstration
(Figure 1) and is most likely associated with the standard error of slope and aspect from
Copernicus DEM given the illumination conditions. This result also demonstrates the
difference between the two methods had the biggest impact for images where $\theta_0$ was high,
resulting in potentially inaccurate retrievals that impact both surface and atmospheric state
variables on relatively mild slopes.

**Table 2.** Image-wide statistics comparing derived properties between the two methods (static
vs. radiance) processing the PRISMA imagery for all three sites.

| Site | PRISMA Method | Mean SSA [$m^2$ $kg^{-1}$] | Mean Broadband Albedo | Mean Liquid water [%] | Mean AOD at 550 nm | Mean water column vapour [mm] |
|---|---|---|---|---|---|---|
| San Juan | Static | 23.3 +/- 14.9 | 0.79 +/- 0.03 | 3.5 +/- 4.8 | 0.05 +/- 0.13 | 6.7 +/- 1.1 |
| | Radiance | 19.6 +/- 5.9 | 0.78 +/- 0.03 | 3.9 +/- 5.0 | 0.01 +/- 0.01 | 6.8 +/- 0.3 |
| Shasta | Static | 11.0 +/- 6.0 | 0.77 +/- 0.04 | 1.6 +/- 3.3 | 0.04 +/- 0.10 | 7.6 +/- 1.3 |
| | Radiance | 10.7 +/- 6.2 | 0.77 +/- 0.05 | 1.9 +/- 3.8 | 0.01 +/- 0.04 | 7.7 +/- 1.1 |
| Toolik | Static | 30.1 +/- 9.6 | 0.85 +/- 0.02 | 0.0 +/- 0.0 | 0.02 +/- 0.03 | 1.0 +/- 0.4 |
| | Radiance | 27.7 +/- 7.9 | 0.84 +/- 0.02 | 0.0 +/- 0.0 | 0.01 +/- 0.01 | 1.0 +/- 0.2 |



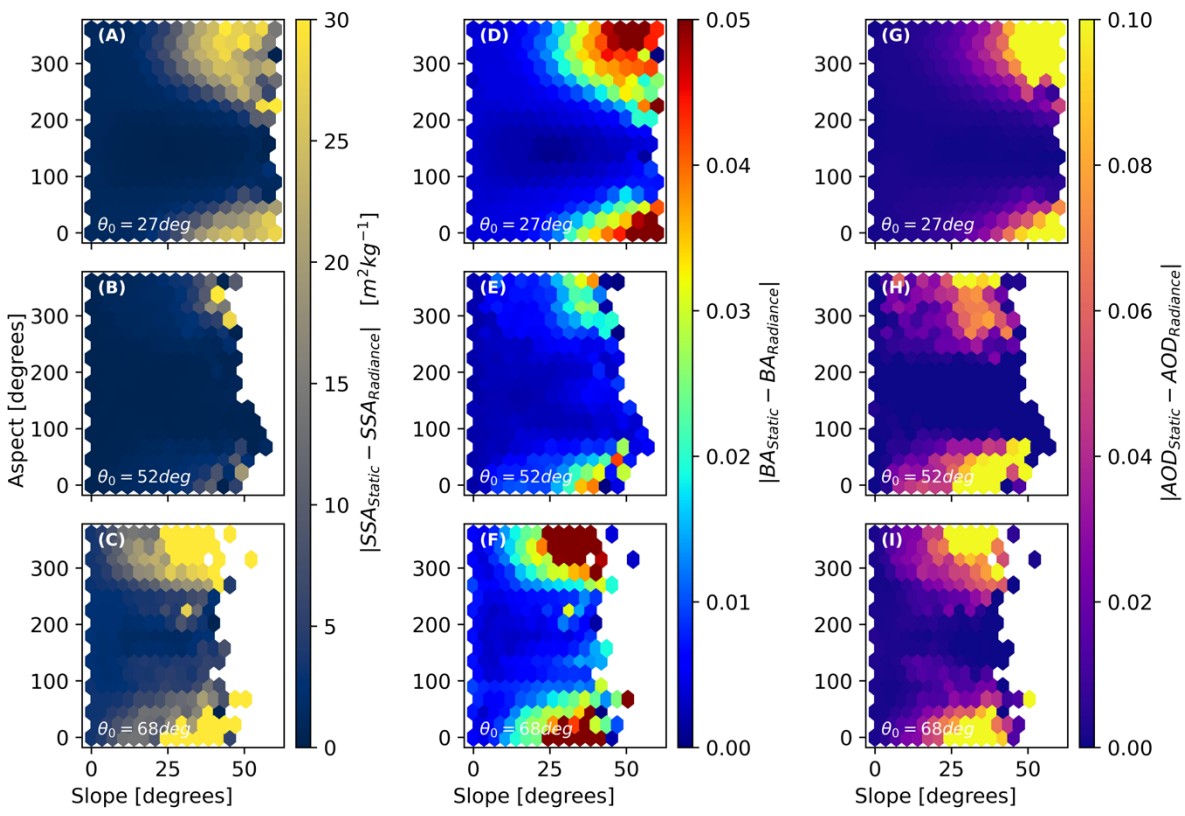


**Figure 6.** 2D Histogram plots showing absolute difference in SSA (left), broadband albedo

(middle) and AOD (left) with respect to slope and aspect across the entire dataset. In this figure

absolute difference is calculated as |Static – Radiance|. This is shown for the San Juan

Mountains site (A,D,G), Shasta site (B,E,H), and Toolik site (C,F,I). The average solar zenith

angle ($\theta_0$) is shown for reference on each panel.


Putting this into spatial context (Figure 7), San Juan site had 37% of pixels (135.3 km$^2$)

with an absolute difference in BA ($|\delta BA|$) >= 0.01 and 14% pixels (49.9 km$^2$) with $|\delta BA|$ >=

0.02. Shasta site had 30% of pixels (16.7 km$^2$) with |δBA| >= 0.01 and 9% pixels (5.1 km$^2$)
with |δBA| >= 0.02. Toolik site had 40% of pixels (325.3 km$^2$) with |δBA| >= 0.01 and 8%
pixels (66.6 km$^2$) with |δBA| >= 0.02.

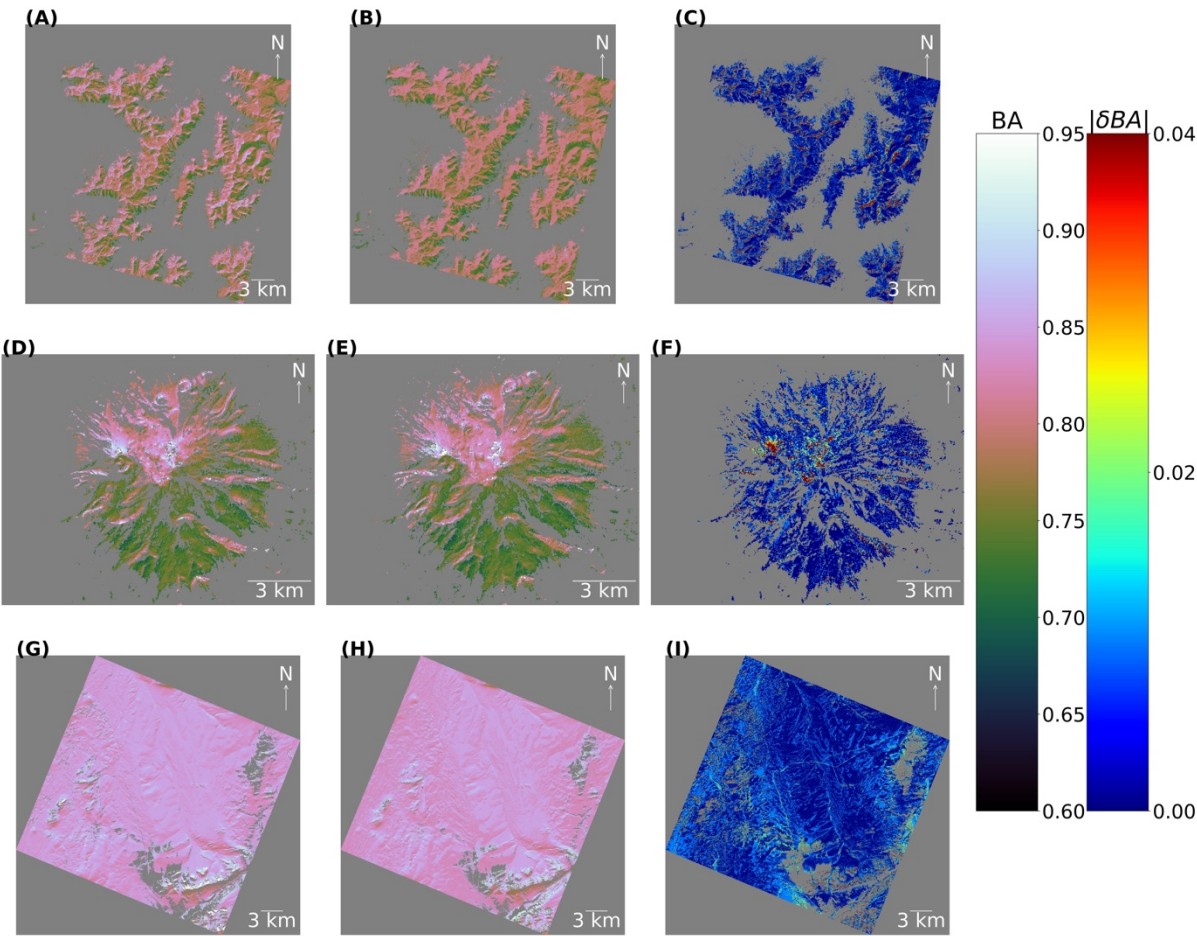

**Figure 7.** Modelled broadband snow albedo (BA) for San Juan Mountains site (A-C), Shasta
Mountain site (D-F), and Toolik site (G-I). Left column represents BA from static method,
middle column represents BA from radiance method, and right column represents absolute
difference in BA ($|\delta BA|$). Dark grey colour symbolizes data that is not a value.

Median $|\delta BA|$ for all sites with respect to $\boldsymbol{\mu_s}$ general increased as $\boldsymbol{\mu_s}$ approached zero

(Figure 8). For example, for the San Juan site, median $|\delta BA|$ ranged from 0.03 to 0.00 across
$\mu_s$. For the Shasta and Toolik sites, median $|\delta BA|$ ranged from 0.02 to 0.00 across $\mu_s$. This
relation was non-linear and depended on the site and illumination conditions. This analysis
demonstrates the levels of uncertainty potentially left in for retrievals relying on static, non-
coincident DEMs. This shows quantitatively the improvements to snow broadband albedo at
30 m scale by using radiance-based approach to be relatively small for well-lit slopes – on the
order 0-1%. While shaded slopes may have errors in snow broadband albedo on the order of
1-3%. Interestingly for the Toolik site, $|\delta BA|$ also increased as $\boldsymbol{\mu_s}$ approached one.

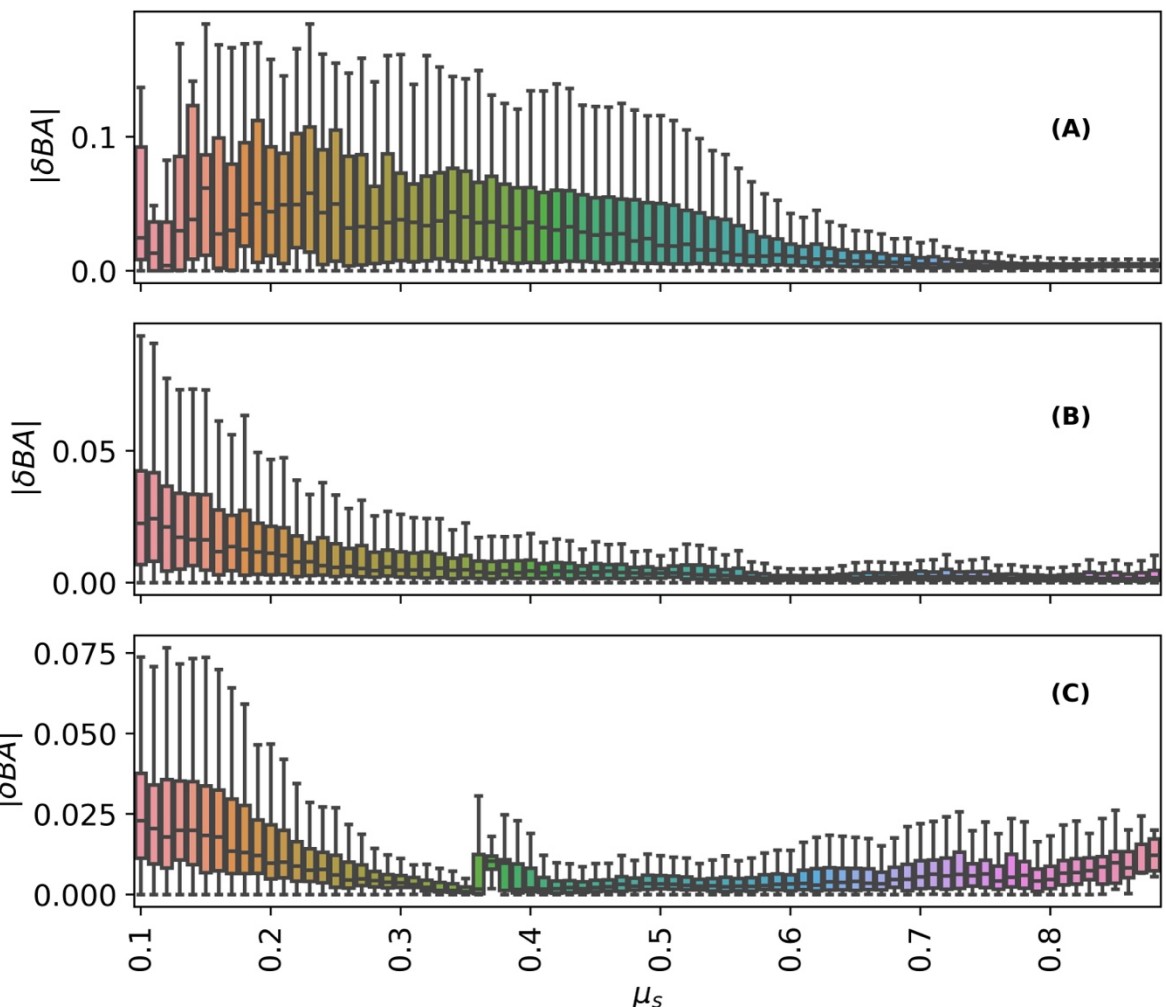

**Figure 8.** Modelled absolute difference in broadband albedo ($|\delta BA| = |BA_{Static} - BA_{Radiance}|$)

for San Juan (A), Shasta (B), and Toolik (C). Note these boxplots were created by rounding

$\mu_s$ to the nearest hundredth place.

## 3.3 Comparing DEM and radiance derived $\mu_s$

At the 30 m pixel scale, Copernicus DEM derived $\mu_s$ had similar overall performance to radiance derived $\mu_s$ (Figure 9), with Copernicus DEM derived $\mu_s$ having slightly higher performance. For example, for the San Juan site, RMSD only varied by 0.006 between the two methods. Similarly, the $R^2$ for Copernicus derived $\mu_s$ was 0.86, while the radiance derived $\mu_s$ was slightly lower at 0.83. This similar overall performance was common amongst the three sites. We found the average bias for radiance derived $\mu_s$ was generally closer to zero (+/- 0.01), and did not show a strong negative or positive direction.

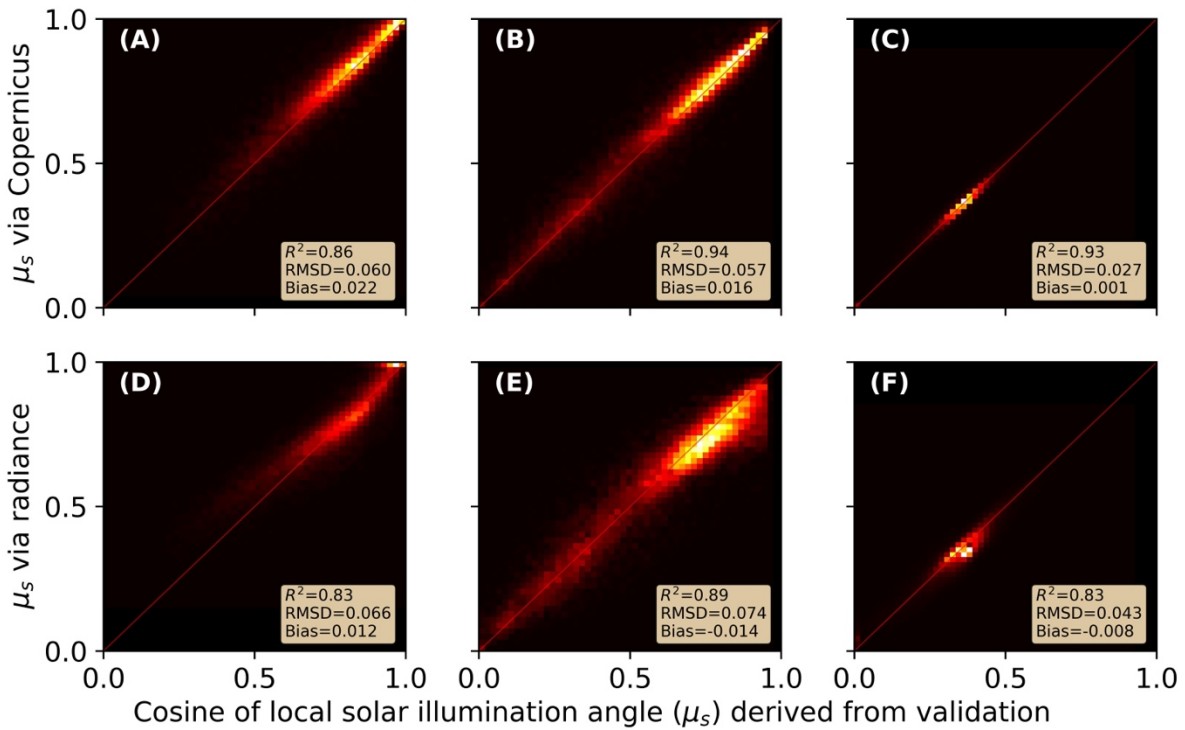

**Figure 9.** Comparing $\mu_s$ at 30 m pixel scale derived from radiance and Copernicus against high resolution DEM for San Juan site (A,D), Shasta site (B,E), and Toolik site (C,F).

**4 Discussion**

**4.1 Radiance derived DEMs may replace coincident DEMs and contain information related to surface roughness**

Derivative slope and aspect maps are prone to errors at 30 m spatial resolutions (Dozier et al., 2022). This is relevant for derived snow products from upcoming missions such as SBG and CHIME which will rely on topographic information to calculate optical properties like snow albedo. These errors can be inherent to the DEM itself, or a product of spatial and/or temporal misalignments (Carmon et al., 2023). Our modelled |δBA| with respect to the non-coincident DEM was similar to work by Donahue et al. (2023), who found slightly higher uncertainties of broadband albedo (ranging from -10 to 10%) for their investigation on Place Glacier, British Columbia, Canada. With the surface and roughness undergoing dramatic change on glaciers throughout a given season, using this radiance-based approach may be especially impactful for improving estimates over glaciers.

Snow surface roughness has long been a challenging issue in modelling snow properties from space where the solar incidence angle at high spatial resolution for snow-on DEM is not well known (Bair et al., 2022). Previous research found radiance derived $\mu_s$ from

airborne imaging spectroscopy showed a negative bias and postulated this could be due to

within-pixel topography, shadows, and surface roughness (Carmon et al., 2023). Since a bi-

directional reflectance function (BRDF) model was not used in their study, it then would be

plausible for the optimal $\mu_s$ to compensate for these effects. Interestingly when using a BRDF

model in our study (i.e., AART) and solving for aspect optimally (therefore informing $\mu_s$, $\mu_v$,

and $\xi$) we did not find a strong bias – negative or positive. Although, we did not take surface

roughness measurements, and therefore do not know to the extent this impacted our study.

Within-pixel shadows, textures, and surface roughness remain difficult to validate, and we

were unable to achieve this in our study. Future work interested in further understanding this

radiance-based approach may investigate how such approaches interact with micro-scale

topography through ground measurements such as terrestrial and airborne lidar.

**4.2 Next steps in possibly improving this radiance-based approach**

While we solved for a few terrain parameters in this study we did not entirely remove

the use of the DEM from the radiance method. The elevation from global DEMs has a much

higher confidence than its derivative products (Dozier et al., 2022). Therefore, we used these

values to inform our atmospheric routine, as well as our shadow casting ray tracing module

(Wilder et al., 2024). Additionally, we used the method presented in Dozier (2022) for

estimating the sky view factor ($V_\Omega$) based on nearby terrain and the pixel itself. This factor

could potentially be problematic but was cited as being not as impactful as $\mu_s$ in propagating
error (Dozier et al., 2022). Therefore, we elected to use $V_\Omega$ derived from the static Copernicus
DEM. However, this could be an area for future improvement, especially in very steep terrain
where $V_\Omega$ becomes small. It is not advised to attempt to add $V_\Omega$ directly into the optimization
routine presented in this study, as it is a function of pixel slope and aspect, and therefore,
altering $V_\Omega$ and aspect together would create invalid solutions. Finally, we used a static value
for slope derived from Copernicus DEM. The slope influences the $\mu_s$ term, but also
influences the passive radiation from nearby slopes. Ultimately, we concluded that aspect had
the largest impact on changing $\mu_s$ (Figure 1), as well as large RMSE reported in previous
work (Dozier et al., 2022; Donahue et al., 2023), and thus was the focus of our study. Caution
is advised in including both slope and aspect together, as non-unique solution space for $\mu_s$
may cause the optimization outputs to become invalid. In summary, elevation, $V_\Omega$, and slope
remain static in our current implementation. Future work may explore other algorithmic
choices to further remove, or improve, static DEM parameters.

Another consideration for improving this method is the inclusion of total column

ozone into the optimization. Previous research has been able to use TOA snow reflectance
data to retrieve reliable estimates of ozone (Kokhanovsky et al., 2021b). In our paper, we
elected for a simpler approach to first investigate the impacts of including terrain in the
optimization. In this paper we input a fixed ozone for the entire image based on coincident
Sentinel-5 measurements. However, it should be stated that ozone impacts a similar spectral
range to $\mu_s$ (Figure 10). It therefore may be beneficial to include ozone in the atmospheric
lookup-table (e.g., MODTRAN, libRadtran) to enable optimization of ozone as well. This
may be beneficial in building more realistic radiance-based methods.

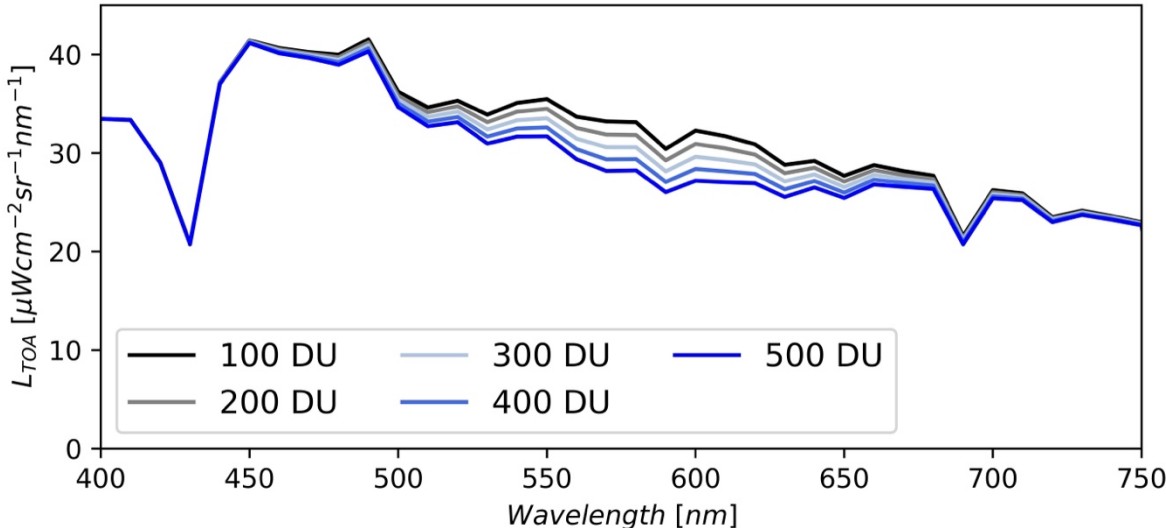


**Figure 10.** Synthetic data showing change in magnitude of top of atmosphere radiance ($L_{TOA}$)
with respect to changing total column ozone for fixed snow surface state variables modelled
with AART, and other fixed atmospheric state variables modelled with libRadtran. Reference
data is based on PRISMA image taken over southern Colorado. Note units of total column
ozone are shown in in Dobson Units (DU).

Finally, future studies should investigate including improvements to BRDF models of
snow (Mei et al., 2022). For example, recent work by Kokhanovsky et al. (2024) has
proposed the use of a two-layer model which may be especially useful for vertically
heterogenous snowpacks. Their method has been tested using EnMAP data and may easily be
transferable to other sensors. The current AART method we used in our paper does not
account for these layers, and instead assumes an optically thick, homogenous snowpack. To
validate both AART, the new layered approach, and future BRDF models, snow pit (i.e.,
vertical profile) measurements of SSA (e.g., Meloche et al., 2023) become essential in
ensuring models accurately account for diverse layering of snow.

**4.3 Big picture implications of the radiance-based approach**
This research responds to the objectives stated in "*Thriving on our changing planet: A*
*decadal strategy for Earth observation from space",* to improve biogeophysical modelling at
scales driven by topography (National Academies of Science, Engineering, & Medicine,
2018), enabling more accurate snow property retrievals in the cryosphere under challenging
illumination conditions. Our work presented on solving terrain where DEM data are not
available, or reliable, may serve to accelerate improvements to satellite remote sensing tools
to monitor and model at both the regional global scale (Sturm et al., 2017), at a critical
juncture in time where northern latitudes are changing fast under a warming climate. This
includes Earth's glaciers, where radiance-based method may have the largest improvements
over static approaches. Our research is complimented by other recent works which show
promise in including terrain in the inversions (Bohn et al., 2024; Bohn et al., 2023; Bair et al.,
2024; Carmon et al., 2023)

We recommend additional coincident AVIRIS-NG flights with spaceborne imaging

spectroscopy datasets to further this work. As we have shown for the San Juan Mountains
site, for particularly warm days, images that are separated by longer than an hour may exhibit
drastically different SSA and liquid water content. As shown in this paper, this creates an
issue when trying to validate improvements to retrieval algorithms.

**5 Conclusion**

In this study we used existing PRISMA L1 TOA imagery to demonstrate the

improvements in modelling snow optical properties when explicitly modelling the terrain in
the inversion. This would especially be true for areas where the surface undergoes rapid
change, such as glaciers. This new method is especially useful for steep mountain terrain
and/or high latitudes where illumination conditions are suboptimal. The $\theta_0$ (solar zenith
angle) was relatively low for the San Juan Mountains site in our study, and thus represents a
lower bound of the improvement in accuracy one could expect. This disparity was
demonstrated further for the Mount Shasta and Toolik sites when $\theta_0$ was larger (i.e. a greater
difference in retrieved properties due to more challenging solar and sensor geometry). Even
for the relatively flat Toolik site, we showed that correctly accounting for incidence angles
can impact snow properties when $\theta_0$ is large. Future work may look to build from this
radiance-based approach to enable better quantification of snow properties at scales impacted
by topography.

*Code Availability.* [https://github.com/cryogars/goshawk](https://github.com/cryogars/goshawk)
*Author contributions.* B.W. created the GOSHAWK algorithm and updates herein, decided
on experiment set-up, and performed the subsequent analysis, as well as being the main
article writer. J.M., J.E. and N.G. provided ideas, comments, and supervised the work.
*Competing interests.* The contact author has declared that neither they nor their co-authors
have any competing interests.
*Acknowledgements.* We acknowledge the Italian Space Agency (ASI) for providing us access
to PRISMA imagery and providing us the foundational data necessary for this research. We
thank Dr. McKenzie Skiles for aiding us in modelling the snow properties from AVIRIS-NG,
and for supplying the dataset.

*Financial support.* This research has been supported by FINESST Award – 21-EARTH21-0249.

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
