# Peer review of "Improved Snow Property Retrievals by Solving for Topography in the Inversion of At-sensor Radiance Measurements"

_EGUsphere, 2024_

## Author Response (AR1)

**R1**

This paper is aimed at the retrieval of atmosphere, snow and underlying terrain properties using imaging spectroscopy. I suggest that the authors make a moderate revision of the manuscript. My comments are given below.

**General comments**

1. GOSHAWK algorithm for the retrieval of snow and atmosphere properties is based on AART and libRadtran. I would suggest that the authors add a section aimed at the description of the accuracy of the algorithm for the parameters listed in Table 1.

   **The atmospheric parameters (water column vapor and aerosol optical depth) are untested in our approach so far.**

   **We discuss this as possible avenues for future work, as well as including ozone in the inversion (per response below). We present a simplified figure where we show that ozone influence may confound terrain signal. And so radiance based methods may look to include ozone in the retrieval for more realistic retrievals.**

2. Could you explain how to do you make the ozone correction. Do not you think that you can retrieve total ozone as well?

   **While it has been shown that ozone may be retrieved, which is an important feature in shorter wavelengths, our focus here is to solve for terrain. This is because retrieving terrain is more pronounced in this spectral range and is a required first step. However, we agree that more accurate ozone estimation is important, and therefore, we have included ozone estimation from Sentinel-5P NRTI O3: Near Real-Time Ozone dataset as input into libRadtran. Future work may expand upon this to also retrieve ozone.**

3. The static and radiance methods give very similar results (inside the retrieval error, see Table 3). Any comment?

**The intention of showing the density histogram was to call out that this method is primarily focused on fixing outlier cases as discussed in Dozier et al. (2022). Therefore, the median and standard deviation give very similar results. Interestingly though, this should promote some amount of confidence in the radiance method, as they give similar average results, despite not using the DEM.**

**Minor comments**

p.2, line 26, please, mention EnMAP

==Done.==

p.3, line 50, degrees?

==This will be revised by "Dozier et al. (2022) found errors in the cosine of the local solar illumination angles ranging from 0.048 to 0.117 (dimensionless) across several sites for Copernicus global DEMs caused by errors in slope and aspect."==

p.7, line 96, could you give Lat/Lon for all sites

==Lat/long are included in Figure 2.==

p.10, line 133, remove 'TBD'

==Done.==

p.11, line 141, 'as PRISMA', add a space

==Done.==

p.13, line 159. According to the ART theory one should write alpha**f and not just alpha in Eq (2). Please, explain.

==This has been corrected in our previous manuscript, and was implemented here in this paper with the correct alpha**f. Corrected code can be found at: https://github.com/cryogars/goshawk/blob/main/scripts/snow.py==

p.14, line 175. Please, check subscripts in Eq.(4).

==We removed EQs (3) and (4) and simply state that the local view angle and local phase angle are functions of terrain. The main idea we try to portray is that optimizing for terrain will impact three different geometry parameters: local view angle, local solar incidence angle, and local phase angle.==

p.16, do not you think that you need to add dust as LAPs and also O3 in Table 1?

**Thank you for this comment. Instead of adding ozone to the inversion, we elected to use a static ozone derived from Sentinel-5P NRTI O3: Near Real-Time Ozone dataset. This serves as forcing into our libRadtran runs for the specific image. We also updated our model to include dust in the inversion. The model can either solve for soot concentration or dust concentration. The dust angstrom parameter and MAE-400 are fixed based from China PM-2.5 tabulated in Caponi 2017 (in a similar fashion to TARTES).**

p.17/18, lines 215, 218, 226, LWC-->liquid water

**We changed instances of LWC to say liquid water.**

p.18, line 225. The accuracy of ART drops in SWIR (bands 1451-1779nm, 1951-2449nm as used by you (Kokhanovsky, Snow Optics, 2021)). Also the band 1951-2449nm is very sensitive to the upper snow layer microphysics (Kokhanovsky, Frontiers in Environmental Science, 2024). This may introduce the biases in your retrievals. You may use the look-up table based on libRadtran to avoid this problem.

**We included the (Kokhanovsky, 2024) in the discussion and specifically point to it as a first attempt at resolving potentially sensitive upper layer microphysics. This would be especially important for areas with very heterogenous snow layering and may be quite beneficial for certain snow environments.**

p.19, line 255 (and p.27, line 304) radiance and static methods give very similar results with variation, which is inside the retrieval error.

**This is correct and we revised to clarify. The intention of showing the density histogram was to call out that this method is primarily focused on fixing outlier cases as discussed in Dozier et al. (2022). Therefore, the median and standard deviation give very similar results. Interestingly though, this should promote some amount of confidence in the radiance method, as they give similar average results, despite not using the DEM.**

p.19, lines 257, 260, 261 - LWC units?

**This was in decimal form before, but to be clearer in our next version we have converted as a percentage with the "%" symbol.**

p.22, could you give average values of the retrieved parameters.

**This is now included in Table 3 .**

**R2**

An important problem in retrieval of remotely sensed variables in mountainous terrain is that the globally available digital elevation models introduce some errors into the analysis. This manuscript proposes a way to approach this issue by solving some of the radiation geometry variables instead of accepting their values calculated from the DEM. Potentially the paper makes an important contribution, but a few issues need to be addressed.

1. The effect of surface roughness on the snow BRDF is not addressed. Although no current snow reflectance model considers this, the roughness contributes to the uncertainty in the analyses and should at least be mentioned.

We have included the discussion of Carmon et al. (2023) into our paper. While we did not conclusively get at surface roughness in this paper, we expand upon this in our discussion section with the following,

"Snow surface roughness has long been a challenging issue in modelling snow properties from space where the solar incidence angle at high spatial resolution for snow-on DEM is not well known (Bair et al., 2022). Previous research found radiance derived $\mu$s from airborne imaging spectroscopy showed a negative bias and postulated this could be due to within-pixel topography, shadows, and surface roughness (Carmon et al., 2023). Since a bi- directional reflectance function (BRDF) model was not used in their study, it then would be plausible for the optimal $\mu$s to compensate for these effects. "

"Interestingly when using a BRDF model in our study (i.e., AART) and solving for aspect optimally (therefore informing $\mu$s, $\mu$v, and $\xi$) we did not find a strong bias – negative or positive. Although, we did not take surface roughness measurements, and therefore do not know to the extent this impacted our study. Within-pixel shadows, textures, and surface roughness remain difficult to validate, and we were unable to achieve this in our study. Future work interested in further understanding this radiance-based approach may investigate how such approaches interact with micro-scale topography through ground measurements such as terrestrial and airborne lidar."

We also included a formal validation assessment of the radiance derived mu_s to mirror analysis done in Carmon et al. (2023) and Dozier et al. (2022).

2. See the comment below about Line 161. A semi-infinite **nonabsorbing** layer of any composition will have a reflectance of 1.0. Albedo = 1-absorption-transmission. If transmission is zero (semi-infinite) and absorption is zero, then Albedo=1.

I have improved the definitions in the BRDF model I am using.

"$r0$ is the reflection function of a semi-infinite non-absorbing snow layer (Tedesco & 163  Kokhanovsky, 2007)."

Please note this r0 term is used to apply directly effects and is only a function of scattering angle.

3. See the comment below about Line 184. An equation that is apparently a crucial component of the analysis is missing.

We did not include our minimization routine which is described in detail in Wilder et al. (2024). We will correct this sentence and do not think it is crucial to include for the story of this paper.

Once these comments are addressed, the paper can be reconsidered for publication. Other comments are included below, along with some suggested references.

Line 13-16. This sentence is missing something, perhaps a "that" following the closing parenthesis in Line 15.

Done.

Line 36. Can eliminate the "off as liquid water."

Done.

Line 69. Figure 1 caption should indicate that aspect is measured clockwise from north, if it is. This is the most common convention, but it's not universal and is in fact inconsistent with a right-hand coordinate system. Sellers' *Physical Climatology* (1965) for example uses aspect 0° south, positive east and negative west. In either case there is a discontinuity at north.

Done.

Line 80. Influenced by viewing geometry **and surface roughness**, which the already cited Bair et al. (2022) show.

Done. We have expanded on the surface roughness in the discussion.

Lines 107-114. You should explain why the AVIRIS-NG data are accurate enough to serve as validation of the PRISMA retrievals. The paragraph mentions the 4 m spatial resolution and 5 nm spectral resolution, but so what? You address this later in the paper by assuming that the 4 m

pixels can be considered a binary (snow or no snow) assessment. But clarify this assumption and identify it as a source of uncertainty. For example, the AVIRIS-NG data in the Indian Himalaya seem to show subpixel snow at that resolution.

**Done.**

Line 133. The Bohn et al. (2024) paper is available as a preprint and should be in the bibliography. The URL is https://papers.ssrn.com/sol3/papers.cfm?abstract_id=4671920.

**Thank you for providing the reference to this paper we have included in the revised copy.**

Line 159. The formulation does not address surface roughness. Does the solution for $\mu_s$ perhaps account for mean of that value over a pixel? Or does fractional shade (Table 1) account?

**Since we used a BRDF model and Carmon et al. (2023) did not, there does not appear to be a negative bias in the radiance derived $\mu_s$ in our paper. We do not know if this simply because the differing use of BRDF vs not, or if simply our limited sites did not contain significantly rough snow. Regardless, we admit to not formally accounting for surface roughness in our paper, and postulate that more ground and airborne coincident lidar campaigns need to happen in SBG/CHIME era that could help answer this very important question you have raised.**

Line 161. Isn't the reflectance of a "semi-infinite **nonabsorbing** snow layer" 1.0? Or is there a better clarification of r0?

**I have improved the definitions in the BRDF model I am using.**

**"$r0$ is the reflection function of a semi-infinite non-absorbing snow layer (Tedesco & 163  Kokhanovsky, 2007)."**

**Please note this r0 term is used to apply directly effects and is only a function of scattering angle.**

Line 168. I don't know if the Van Rossum citation is necessary. The atan2 function has been available for decades in most computer languages, before most Python coders were born.

**We have removed the reference.**

Line 170. What does "circular" mean in this context? Aspect is discontinuous at north.

**Thank you we have clarified how we define aspect.**

Line 184. The "following equation" seems to be missing.

**It was not a critical piece of this paper, and it was just showing our minimization problem we solve (see Wilder et al. (2024)). This paper used to have this equation but we decided to remove it to keep the paper more focused. And so some text got left behind. We apologize for the confusion.**

Table 1. The caption does not explain the meaning of the three colors in the table. The range of LAP concentration (ng/g) ranges from 0 to 0.5e-5. This can't be correct; the maximum dimensionless mass concentration would be 0.5e-14.

**The colors are removed as per the editor's comments. The range of soot concentration was displayed incorrectly in our table and was not scaled properly after optimization. Per the other reviewer we will remove soot concentration and instead model as dust. We have modeled dust in units of micro grams / gram.**

Line 208. Rob Green and I agree that "higher" spatial resolution is ambiguous and suggest "finer" instead.

**Thank you for the insight and used the term "finer" instead here.**

Line 227. The citation to a 43-year-old thesis (Segelstein, 1981) is unusual. Unless there's an important, peer-reviewed, published update, I suggest citing Hale and Querry (1973).

**We cited Hale and Querry (1973) instead as recommended by the reviewer.**

Line 398. The correct year is 2018.

**Thank you for providing the correct references below.**

Line 410. The Dozier-Frew (1981) paper does not address the view factor. I think you mean Dozier and Frew (1990), but I would recommend instead Dozier (2022) which addresses the issue where the pixel slope itself is a significant part of the view factor calculation. That code is available on the MATLAB file exchange (https://www.mathworks.com/matlabcentral/fileexchange/94800-topographic-horizons).

**See, "CC3: 'Reply on RC2, Line 410', Brent Wilder, 18 Jul 2024". In short, we have adapted the Dozier (2022) method to include impacts from pixel itself. We apologize for mistakenly referencing the Dozier-Frew (1981) paper, we intended to reference the 1990 paper. However, now we will reference the 2022 paper.**

Line 471. "Chime" should be in all upper case, "CHIME."

**Typo fixed.**

Line 449. The references should include their DOIs, mostly available and making the citation much easier to find if the reader wants to.

**All citations and DOI will be reviewed and included as available.**

Line 457. The "author" of this publication is "National Academies of Science, Engineering, and Medicine." The correct citation is shown below, including the DOI.

**Thank you!**

Lines 503 & 529. The citation to "McKenzie Skiles, S." should instead be "Skiles, S.M." as is correct in other citations in the bibliography.

**Indeed it should. We have used the proper format for all references now in this draft.**

References mentioned in the Review

Dozier, J.: Revisiting topographic horizons in the era of big data and parallel computing, IEEE Geoscience and Remote Sensing Letters, 19, 8024605, doi: 10.1109/LGRS.2021.3125278, 2022.

Dozier, J. and Frew, J.: Rapid calculation of terrain parameters for radiation modeling from digital elevation data, IEEE Transactions on Geoscience and Remote Sensing, 28, 963-969, doi: 10.1109/36.58986, 1990.

Hale, G. M. and Querry, M. R.: Optical constants of water in the 200-nm to 200-μm wavelength region, Applied Optics, 12, 555-563, doi: 10.1364/AO.12.000555, 1973.

National Academies of Sciences, Engineering, and Medicine: Thriving on Our Changing Planet: A Decadal Strategy for Earth Observation from Space, National Academies Press, Washington, DC, 716 pp., doi: 10.17226/24938, 2018.